# Health, social, and economic characteristics of patients enrolled in a COVID-19 recovery program

**Suzanne M. Simkovich** [1,2], **Naheed Ahmed**[1], **Jiling Chou**[1‡], **Asli McCullers**[1‡], **Eric M. Wisotzky**[3], **Jennifer Semel**[3‡], **Kathryn Pellegrino**[3], **Derek DeLia**[1,4‡], **William S. Weintraub**[1,5]

**1** MedStar Healthcare Delivery Research Network, MedStar Health Research Institute, Hyattsville, MD, United States of America, **2** Division of Pulmonary and Critical Care Medicine, Georgetown University School of Medicine, Washington, DC, United States of America, **3** Division of Rehabilitation Medicine, MedStar National Rehabilitation Network, Washington, DC, United States of America, **4** Department of Plastic and Reconstructive Surgery, Georgetown University School of Medicine, Washington, DC, United States of America, **5** Department of Medicine, Georgetown University School of Medicine, Washington, DC, United States of America

യ These authors contributed equally to this work.
‡ JC, AM, JS and DD also contributed equally to this work.
* suzanne.m.simkovich@medstar.net

**Data Availability Statement:** All relevant data are within the paper and its Supporting Information files.

## Abstract

At least one in five people who recovered from acute COVID-19 have persistent clinical symptoms, however little is known about the impact on quality-of-life (QOL), socio-economic characteristics, fatigue, work and productivity. We present a cross-sectional descriptive characterization of the clinical symptoms, QOL, socioeconomic characteristics, fatigue, work and productivity of a cohort of patients enrolled in the MedStar COVID Recovery Program (MSCRP). Our participants include people with mental and physical symptoms following recovery from acute COVID-19 and enrolled in MSCRP, which is designed to provide comprehensive multidisciplinary care and aid in recovery. Participants completed medical questionnaires and the PROMIS-29, Fatigue Severity Scale, Work and Productivity Impairment Questionnaire, and Social Determinants of Health surveys. Participants (n = 267, mean age 47.6 years, 23.2% hospitalized for COVID-19) showed impaired QOL across all domains assessed with greatest impairment in physical functioning (mean 39.1 ± 7.4) and fatigue (mean 60.6 ±. 9.7). Housing or "the basics" were not afforded by 19% and food insecurity was reported in 14% of the cohort. Participants reported elevated fatigue (mean 4.7 ± 1.1) and impairment with activity, work productivity, and on the job effectiveness was reported in 63%, 61%, and 56% of participants, respectively. Patients with persistent mental and physical symptoms following initial illness report impairment in QOL, socioeconomic hardships, increased fatigue and decreased work and productivity. Our cohort highlights that even those who are not hospitalized and recover from less severe COVID-19 can have long-term impairment, therefore designing, implementing, and scaling programs to focus on mitigating impairment and restoring function are greatly needed.

**Funding:** This project did not have assigned funding for its completion. Suzanne M. Simkovich was supported by funding from the National Heart, Lung, and Blood Institute K12HL137942.The funders had no role in study design, data collection and analysis, decision to publish, or preparation of the manuscript.

**Competing interests:** The authors have declared that no competing interests exist.

## Introduction

The continued spread of the coronavirus-2 (SARS-CoV-2) has resulted in millions of people worldwide being infected and developing symptoms of the coronavirus disease 2019 (COVID-19), with the expectation of millions more being infected [1]. The spectrum of resulting disease has been quite varied from mild asymptomatic illness to prolonged hospitalization and debility, to death [2–4]. For those who survive, many recover without any symptoms, but at least one in five experience continued illness following the acute phase of infection [3]. This group, who continue to have ongoing symptoms following the acute illness period, is characterized by the Centers for Disease Control and Prevention with the diagnosis of post-acute sequalae of SARS-COV2 infection (PASC) [4, 5]. PASC encompasses symptoms of profound fatigue, muscle aches and pains, dyspnea, myalgia, cognitive deficits, and depression that continue four or more weeks following infection [3, 5, 6]. These symptoms are similar to the ongoing symptoms experienced by prior coronavirus pandemics in patients who were infected with severe acute respiratory syndrome coronavirus (SARS-CoV) and Middle East respiratory syndrome coronavirus (MERS-CoV) resulting in chronic illness and debilitation [7, 8].

Chronic illnesses are well known to have negative mental and physical consequences resulting in a lower quality- of- life (QOL), loss of productivity and financial hardships, along with adverse psychosocial effects [9, 10]. Patients who developed chronic illness from ongoing symptoms following SARS-CoV and MERS-CoV showed reduced QOL and productivity along with impairment in the ability to work [11, 12]. However, recovery programs designed to mitigate these consequences demonstrated results in improving these factors [7, 13]. As of late 2021, there is limited information on the psychologic, social, and economic well-being among patients who have recovered from acute SARS-CoV2, but develop PASC.

As our country and the world are facing many patients who have been infected with the SARS-CoV-2 virus, there is an urgent need to not only understand the physical and mental health consequences of COVID-19, but the QOL and socio-economic impacts of COVID on populations. Understanding these consequences will provide vital information to inform the design, implementation, and scaling of programs to mitigate the negative consequences and restore patients to prior health status and function. To assist in the recovery of patients with PASC, MedStar Health established a multidisciplinary the MedStar COVID Recovery Program (MSCRP), which is a comprehensive multidisciplinary program directed at connecting patients with resources to manage and mitigate PASC. We assessed the mental and physical symptoms, QOL, fatigue, socio-economic status, and work and productivity of patients who enrolled in this program and report our results in this manuscript. We hypothesize that those who have continued mental and physical symptoms following acute COVID-19 will have a lower QOL, elevated fatigue, decreased work and productivity, and socio-economic hardships.

## Materials and methods

### Program description and participant selection

The MedStar Health COVID Recovery Program (MSHCRP) is housed in the MedStar Health National Rehabilitation Institute. MedStar Health serves 8.8 million patients across the greater Washington, DC, central and southern Maryland, and northern Virginia area. MSHCRP is a rehabilitation program led by physical medicine and rehabilitation physicians and advanced practice providers who specialize in the rehabilitation of patients with a variety of physical impairments. Admission to the recovery program is limited to patients at least six weeks post-start of COVID symptoms, and either a documented positive COVID-19 test or physician referral stating that the patient's symptoms indicated COVID-19 without a positive test.

Patients could enroll in the program regardless of if he or she was a prior MedStar patient or not.

Patients were asked to complete a series of surveys prior to the first appointment (survey instruments available in S1 File). Patients were asked to report demographic information, including age, sex, race, ethnicity, and health insurance type. All responses were self-reported. Response rates varied by questionnaires. Patients are then seen in-person or by telehealth to accommodate transportation barriers and prevent the spread of COVID-19. At this initial evaluation process, necessary referrals are made to medical specialists to address persistent physical and mental symptoms. Patients are assigned a patient navigator and community health advocate to assist with scheduling and managing care, and social service referrals. The social service referral component of the clinic is intended to address social determinants of health, such as assisting patients experiencing challenges with working, housing payments, or purchasing food. Patients are scheduled and reminded of repeated follow-ups every two months with the program for assessment, recommendations, and referrals.

## Ethics approval and manuscript standards

The study protocol was reviewed and approved by the institutional review board of MedStar Health Research Institute (STUDY00003880) and consent was waived as an exempt protocol. As this is a cross-sectional study, items are reported in accordance with the STROBE Statement (see S1 File) [14].

## Instruments

Medical Questionnaire: Participants were asked questions regarding medical history, current symptoms, and COVID-19 related hospital admissions.

Patient-Reported outcomes Measurement Information System (PROMIS) ® 29: The PROMIS-29 v2.0 profile assesses mental and physical QOL using a single 0–10 numeric rating scale for seven health domains (physical function, fatigue, pain interference, pain intensity, depressive symptoms, anxiety, ability to participate in social roles and activities, and sleep disturbance) [15]. Scores for each health domain are averaged and interpreted based on the PROMIS® T-Scores which is a mean score of 50 and a standard deviation of ten [15]. The PROMIS survey is validated for use to assess impacts of health care interventions and track changes in health over time [15].

Fatigue Severity Scale (FSS): The FSS is a unidimensional, nine-item questionnaire measuring the severity of fatigue [16]. Participants rate 9-items on a 7-point Likert scale based on the previous week. Responses to this scale ranged from zero to ten with higher scores reflecting lower fatigue. Mean scores are disease and population-specific and the survey is validated in healthy subjects and different disorders known to be commonly associated with fatigue [17].

Social Determinants of Health Survey (SDOH): The Aunt Bertha social determinants of health survey is an eight question survey designed to measure the economic and social conditions that influence the health and QOL of individuals [18]. Items were answered using "yes" and "no" options. SDOH is validated for use in community health settings [19].

Work Productivity and Activity Impairment Scale (WPAI-SHP): The WPAI- SHP is a six question instrument used to measure impairments in both paid and unpaid work, absenteeism, presenteeism as well as the impairments in unpaid activity over the past seven days [20]. WPAI outcomes are expressed as impairment percentages, with higher numbers indicating greater impairment and less productivity [20]. The WPAI is validated across a variety of diseases including asthma and allergic rhinitis and can be utilized for specific conditions, such as COVID-19 [20] We utilized this survey to ask specifically about COVID-19.

## Statistical analysis

Sample size calculations were not performed *a priori* as our analysis is exploratory in nature. All surveys were scored per the parameters of the survey [15–20]. Missing data was handled per the parameters of the survey. Summary statistics included count and percentage of sample, mean and standard deviation. Normality was tested using D'Agostino-Pearson test and median values and interquartile ranges were calculated for the non-normally distributed sub-groups. We conducted a subgroup analysis by age ($\geq 50$ years or $< 50$ years), race/ethnicity (Asian, Black, Hispanic, White, unknown or other), gender (male or female), medical diagnosis of asthma (yes or no) or allergies (yes or no) or diabetes (yes or no) and hospitalized for COVID-19 (yes or no) in relation to PROMIS scores [21]. Two-sided Student's t-test was utilized if the PROMIS variables did not reject normality and two-sided Wilcoxon rank sum test were performed for non-normal distributions to compare differences between groups of each selected characteristics. Statistical significance was set at p-value less than 0.05. All analyses were conducted using R software version 4.0.5 (Shake and Throw) using the packages tableone and moments [22].

## Results

### Patient demographics

Between December 7, 2020 and July 6, 2021, 267 patients were enrolled in MSCRP and agreed to complete the intake questionnaires (Table 1). The majority of patients were female (76.8%), with a racial/ethnic distribution of 55.4% white, 32.6% Black/ African American, 4.1% Asian, and 3% Hispanic. The mean age of our participants was 47.6 years old with 45.7% being married. All except two participants of the 267 enrolled were insured, with the majority carrying a private policy as part of their insurance coverage (n = 246). Less than a quarter of the cohort had public insurance-Medicare (n = 28) and Medicaid (n = 48). This proportion likely reflects the employment status where 72.7% reported full time employment, 7.1% retired and 4.1% employed part time. In this cohort 9.4% were unemployed prior to diagnosis and 2.2% recently unemployed. Employment varied across industries with the greatest number of patients working in the business or non-profit sector (15.0%) and medical (12.4%). Most patients were referred to the MSCRP by their primary care provider (64.8%) or self-referred (11.2%). Twenty-eight percent, 33%, and 38% of patients had COVID-19 within 6 weeks to 3 months, 3 months to 6 months, greater than 6 months, respectively, prior to enrollment in MSHCRP.

### Participant medical history and symptoms at program enrollment

Patients' symptoms varied considerably with some being much more common than others (Table 2). The most reported symptoms were fatigue (70.8%), sleep disturbance (47.9%), body aches or pains (40.4%), and general weakness (34.5%). Other commonly reported symptoms were respiratory-related: shortness of breath (50.2%); neurologic: headache (47.9%), difficulty focusing (47.6%), lightheadedness or dizziness (39.3%), feeling down or tearful or depressed (31.1%), impaired memory (37.5%); and cardiovascular: chest pain (30.3%), palpitations or heart racing (34.8%). Four patients did not report any symptoms (1.5%). Patients also varied widely in their prior medical history. One third of the patients (30.9%) reported no prior medical history. Allergies were the most common medical diagnosis (38.6%) followed by asthma (21.7%). Only 8.8% of the cohort had diabetes and 4.8% had cardiovascular disease, both of which have been correlated with negative outcomes in COVD-19 [23–25]. Twenty-three percent of patients had been hospitalized for COVID-19 or close to their COVID-19 diagnosis which can serve as an indicator of severe COVID [4]. Of this group who reported

**Table 1. Patient demographics (n = 267).**

| | | |
|---|---|---|
| **Sex, n (%)** | | |
| | Female | 205 (76.8%) |
| | Male | 62 (23.2%) |
| Race/Ethnicity, n (%) | | |
| | Black/AA | 87 (32.6%) |
| | Hispanic | 8 (3.0%) |
| | Asian | 11 (4.1%) |
| | White | 148 (55.4%) |
| | Unknown | 13 (4.9%) |
| Age, mean (s.d.) | | 47.6 (12.9) |
| Health Insurance, n * | | |
| | Private | 246 |
| | Medicaid | 48 |
| | Medicare | 28 |
| | Self-Pay | 2 |
| | Worker's Compensation | 2 |
| Employment Status, n (%) | | |
| | Full time | 194 (72.7%) |
| | Part time | 11 (4.1%) |
| | Recently Unemployed | 6 (2.2%) |
| | Retired | 19 (7.1%) |
| | Student | 6 (2.2%) |
| | Unemployed | 25 (9.4%) |
| | Not applicable | 6 (2.2%) |
| Occupation, n (%) | | |
| | Medical | 33 (12.4%) |
| | Engineering | 3 (1.1%) |
| | Legal | 6 (2.2%) |
| | Government Services | 16 (6%) |
| | Education | 13 (4.9%) |
| | Scientific/ Technical | 14 (5.2%) |
| | Student | 6 (2.2%) |
| | Business/ Non-Profit | 40 (15.0%) |
| | Retail | 5 (1.9%) |
| | Other | 78 (29.2%) |
| | Unknown | 53 (19.9%) |
| Religion, n (%) | | |
| | Catholic | 38 (14.2%) |
| | Protestant | 89 (33.3%) |
| | Jewish | 5 (1.9%) |
| | Hindu/ Buddhist | 5 (1.9%) |
| | Other | 22 (8.2%) |
| | No religious preference/Unknown | 108 (40.4%) |
| Marital Status, n (%) | | |
| | Single | 107 (40.1%) |
| | Married | 122 (45.7%) |
| | Divorced | 24 (9%) |
| | Widowed | 7 (2.6%) |

(*Continued*)

**Table 1.** (Continued)

| Sex, n (%) | | |
|---|---|---|
| | Living with partner | 4 (1.5%) |
| | Separated/ not applicable | 3 (1.1%) |
| COVID-19 Management, n (%) | | |
| | Hospitalization | 62 (23.2%) |
| | Intensive Care Unit | 6 (2.2%) |
| | Ventilator | 1 (<1%) |
| | Re-hospitalized or ER visit following COVID-19 initial hospitalization | 57 (21.3%) |
| Referral to Clinic, n (%) | | |
| | Self | 30 (11.2%) |
| | Primary Care Physician | 173 (64.8%) |
| | Hospital discharge | 17 (6.4%) |
| | Nurse or case manager | 3 (1.1%) |
| | Other/ unknown | 44 (16.5%) |
| Time since COVID-related symptoms began, n (%)** | | |
| | 6 weeks to 3 months | 32 (28.8%) |
| | 3 months to 6 months | 37 (33.3%) |
| | Greater than 6 months | 42 (37.8%) |

*Counts do not add to n = 267 as participants may have more than one type of insurance.

** 111 patient responded to this question (n = 140)

hospitalization, six participants reported being admitted to an ICU (2.2%) and only one reported being on a ventilator.

## Participant QOL, fatigue, work and productivity, and socio-economic assessments

Participants showed reduced QOL on the PROMIS-29 with mean scores showing worse than normal QOL in all domains of the survey (Table 3). The mean scores were not within one standard deviation of normal (mean of 50 ± 10) in the domains of physical functioning (mean 39.1 ± 7.4) and fatigue (mean 60.6 ±. 9.7). All other mean scores were within one standard deviation. The mean score on the fatigue severity scale (mean 4.7 ± 1.1) showed increased fatigue compared to the mean scores of the general population [17, 26].

On our assessment of social determinants of health, a fraction of participants reported issues with meeting basic needs. Forty-seven participants (19.3%) reported not being able to pay for rent or mortgage or pay for the basics. Thirty-nine participants (16.0%) worry about food insecurity, and 34 (13.9%) reported it is hard to get and keep a job. On the WPAI, 63% of our population reported activity impairment, 61% reported impaired work productivity, 56% reported impaired presenteeism, and 34% reported absenteeism. Of note, only 153 participants completed the WPAI and this does not reflect the full sample of 267 participants.

## Patient subgroup analysis

A subgroup analysis was performed to assess the demographic factors and medical conditions that may lead to impairment in QOL on the PROMIS-29 by domain (see online supplement for full analysis). We found small differences in participants <50 years of age having greater anxiety than those >50 years of age, along with greater sleep disturbance among those who are of Hispanic or other ethnic origin, and greater pain intensity and fatigue amongst females

**Table 2. COVID-19 health metrics.**

| Current Symptoms (n = 267) | | n (%) |
|---|---|---|
| | Fever | 24 (9.0%) |
| | Chills | 32 (12%) |
| | Fatigue | 189 (70.8%) |
| | Sleep Disturbance | 128 (47.9%) |
| | Loss of appetite | 44 (16.5%) |
| | Unexpected change in weight | 39 (14.6%) |
| | Blurry Vision | 55 (20.6%) |
| | Sensitivity to Light | 41 (15.4%) |
| | Light-headedness or dizziness | 105 (39.3%) |
| | Headache | 133 (49.8%) |
| | Loss or decrease in smell | 58 (21.7%) |
| | Runny nose | 25 (9.4%) |
| | Sinus congestion | 66 (24.7%) |
| | Sore Throat | 30 (11.2%) |
| | Hoarse Voice | 35 (13.1%) |
| | Chest pain | 81 (30.3%) |
| | Palpitations or heart racing | 93 (34.8%) |
| | Cough | 66 (24.7%) |
| | Shortness of breath | 134 (50.2%) |
| | Abdominal Pain | 42 (15.7%) |
| | Nausea and/or vomiting | 44 (16.5%) |
| | Change in bowel habits | 51 (19.1%) |
| | Urinary changes | 19 (7.1%) |
| | Body aches or pains | 108 (40.4%) |
| | Weakness or numbness in a specific body area | 53 (19.9%) |
| | General weakness | 92 (34.5%) |
| | Tremors | 17 (6.4%) |
| | Skin, hair or nail changes | 53 (19.9%) |
| | Confusion, disorientation | 59 (22.1%) |
| | Impaired memory | 100 (37.5%) |
| | Difficulty focusing | 127 (47.6%) |
| | Feeing down, tearful and/or depressed | 83 (31.1%) |
| | Feeling anxious or fearful | 102 (38.2%) |
| | None of the above | 4 (1.5%) |
| **Medical History** (n = 267) | | **n (%)** |
| | Heart disease | 13 (4.8%) |
| | Diabetes | 24 (8.8%) |
| | Allergies | 105 (38.6%) |
| | Eczema | 28 (10.3%) |
| | Asthma | 59 (21.7%) |
| | Lung Disease | 2 (0.7%) |
| | Kidney Disease | 6 (2.2%) |
| | Cancer | 15 (5.5%) |
| | Autoimmune Disease | 17 (6.3%) |
| | None | 84 (30.9%) |

*267 participants completed these questions. Participants could select more than one symptom or medical history diagnoses.

**Table 3. Survey responses.**

| PROMIS-29 (n = 245) | | Score Interpretation |
|---|---|---|
| Physical Function, *mean (s.d.)* | 39.1 (7.4) | Survey is calibrated to a mean score of 50 with a standard deviation of 10. For physical function and ability to participate mean scores >50 are better than normal. All other domains mean scores <50 are better than normal. * Pain intensity score is 0–10 and is not a domain based on a T-score. |
| Anxiety, *mean (s.d.)* | 58.7 (9.8) | |
| Depression, *mean (s.d.)* | 54.0 (9.6) | |
| Fatigue, *mean (s.d.)* | 60.6 (9.7) | |
| Sleep Disturbance, *mean (s.d.)* | 55.7 (8.9) | |
| Ability to Participate, *mean (s.d.)* | 42.1 (8.8) | |
| Pain Interference, *mean (s.d.)* | 57.1 (10.3) | |
| Pain Intensity, *mean (s.d.)** | 4.2 (2.8) | |
| **Fatigue Severity Scale** (n = 247) | | |
| Fatigue Severity Scale, *mean (s.d.)* | 4.7 (1.1) | Higher score, greater severity of fatigue. Normal scores in healthy subjects have been measured in the following mean score 3.0 ±1.08 (Valko et al 2008), mean score 2.3 ± 0.7) (Grace et al 2006) |
| **Social Determinants of Health** (n = 244) | | |
| Unable to pay for basics, *n (%)* | 47 (19.3%) | Number of participants who answered yes to the question. |
| Worry food might run out, *n (%)* | 39 (16.0%) | |
| Food didn't last, *n (%)* | 29 (11.9%) | |
| Not able to pay for rent or mortgage, *n (%)* | 47 (19.3%) | |
| Nowhere to live, *n (%)* | 15 (6.1%) | |
| No access to transportation, *n (%)* | 13 (5.3%) | |
| Hard to get and keep a job, *n (%)* | 34 (13.9%) | |
| Utilities threatened to be turned off, *n (%)* | 15 (6.1%) | |
| **Work Productivity & Activity Impairment** (n = 153) | | |
| Absenteeism (work time missed), *mean %* | 34% | Outcomes are expressed as percentages with higher numbers indicating greater impairment and less productivity. |
| Presenteeism (impairment at work/ reduced on-the-job effectiveness), *mean %* | 56% | |
| Work Productivity (overall work impairment/ absenteeism plus presenteeism), *mean %* | 61% | |
| Activity Impairment, *mean %* | 63% | |

compared to males. Those patients who had underlying asthma had scores that were worse than those without asthma in the domains of fatigue and ability to participate in social roles and activities. Those who were hospitalized for COVID-19 had increased impairment in physical functioning compared to those who were not hospitalized. There were no differences in whether a patient had allergies or diabetes.

## Discussion

Our cohort of patients with PASC show impaired QOL across all domains, increased fatigue, impaired work and productivity along with increased social needs and financial hardships. The clinical symptoms in this cohort of patients with PASC reports were consistent with the symptoms of other cohorts with PASC and similar clinical symptoms and impairment to cohorts who suffered from SARS and MERS [7, 8, 11, 12, 27–31]. Notably, our cohort had a from milder cases of COVID-19 than other cohorts based utilizing hospitalization as a surrogate, as slightly less than one quarter were hospitalized and only 2.2% report being admitted to an intensive care unit, meaning our cohort was less likely to have had severe COVID-19 [32]. Further, our cohort is young with a mean age of 48 years old and a high level of private insurance and employment. Despite our cohort being less ill, young and employed, there was significant impairment as we found in this analysis. Based on the limits of this investigation, we cannot conclude this is due to PASC, but indicates or cohort has social and financial impairments. Our findings indicate that recovery programs focused on PASC need to address factors associated with QOL, fatigue, and socio-economic hardships, and include patients with milder disease.

To our knowledge, this is the largest cohort to characterize the QOL, fatigue and work, socio-economic characteristics, and productivity of patients suffering from PASC. Most studies have focused on evaluating the physical, mental, and clinical findings of patients who have PASC with less focus on QOL and socioeconomic outcomes. There are a few cohort studies that have cross-sectionally evaluated the QOL of patients with PASC and have shown mixed results. Carfi et al reported 44% of 143 patients with PASC had impairment of QOL on the EuroQOL-5D Visual Analogue Scale [33]. Jacobs et al. reported impaired QOL in 39.8% of participants (n = 183) with PASC as measured by the PROMIS® Scale v1.2- Global Health [30]. Studies that evaluated all patients who were diagnosed with COVID-19 and assessed QOL following recovery are heterogenous, which is most likely due to these studies including all patients and not just those with PASC [28, 31, 34]. The patients with PASC who enrolled in the Mayo Clinic's Activity Rehabilitation Program are similar to our cohort [27]. The majority of the Mayo patients were women (68%), mean age of 45 years old, and only 25% had been hospitalized for COVID-19. Similar clinical symptoms to our cohort were reported. Both the Mayo cohort and ours showed impairment in work productivity with the Mayo cohort showing only 46% of participants returning to unrestricted work and in our cohort only 39% reported no impairment in returning to work [27]. At this time, the majority of the published studies on PASC cohorts are heterogeneous, making definitive conclusions difficult to draw [35].

There are significant strengths to this evaluation. To our knowledge, this is the largest cohort of patients with PASC where the dimensions of QOL, fatigue, work and productivity along with social determinants and financial hardships were measured [35]. Second, while our PASC cohort had a less severe course of COVID-19, there was impairment in key domains of health and function, thus drawing attention to complex health needs of patients who had mild to moderate disease. Third, we have a racially and ethnically diverse sample that can be generalizable across the United States.

Despite the strengths of this evaluation, there are limitations. First, this is a cross-sectional analysis on a self- selected population. This limited the assessment of symptoms to one point in time. Patients had to either self-refer or communicate to a clinical provider or case manager continued symptoms following acute COVID-19 and agree to enroll in a program to mitigate these symptoms. This may have created self-selection bias in our sample as our patients recognize their ongoing symptoms and desired to improve their clinical status, and not include all

patients who were diagnosed with COVID-19. Further, our enrolled sample is predominantly female, which is not uncommon for ongoing symptoms following a post-viral syndrome and engagement in healthcare [27, 36]. Second, we administered surveys prior to the first visit in the clinic and responses are self-reported, thus creating a potential social desirability bias. Third, although racially and ethnically diverse, our cohort encompasses one population in the Washington, D.C. area and may not reflect the health and care seeking behaviors of other populations. Fourth, we do not have baseline measurements on our cohort prior to infection with COVID-19. Therefore, we are unable to assess distinctly if our findings are significantly different from baseline. Fifth, we did not distinguish our population based on variants that caused the initial infection, which could lead to different clinical outcomes [35]. Unfortunately, our population was not tested for particular variants making this differentiation not possible. There may be different phenotypes of PASC symptoms based on variant of infection, but this has yet to be elucidated. Lastly, we are assessing ability to work in our population, but we do not know the true disability status of our patients which may impact the number of patients who are able to work.

## Conclusions

Patients with PASC following acute COVID-19 have continued mental and physical symptoms with resulting fatigue, impairment in QOL, socioeconomic hardships, and decreased work and productivity. Our cohort highlights that even those who recover from less severe courses of COVID-19 can have impairment. Understanding these factors can inform the design, implementation, and scale-up of effective interventions to mitigate these ongoing consequences and attempt to restore patients to prior functioning and status. Ongoing research is needed to further elucidate the specific socioeconomic needs of populations following an acute infection with COVID-19.

## Supporting information

**S1 File. "Health, social, and economic characteristics of patients enrolled in a COVID-19 recovery program".**
(DOCX)

**S1 Data.**
(CSV)

**S2 Data.**
(CSV)

**S3 Data.**
(CSV)

**S4 Data.**
(CSV)

**S5 Data.**
(CSV)

**S6 Data.**
(CSV)

**S7 Data.**
(CSV)

## Author Contributions

**Conceptualization:** Suzanne M. Simkovich, Naheed Ahmed, Eric M. Wisotzky, Jennifer Semel, Derek DeLia, William S. Weintraub.

**Data curation:** Suzanne M. Simkovich, Eric M. Wisotzky, Jennifer Semel.

**Formal analysis:** Suzanne M. Simkovich, Naheed Ahmed, Jiling Chou, Derek DeLia.

**Funding acquisition:** Suzanne M. Simkovich, William S. Weintraub.

**Investigation:** Naheed Ahmed, Eric M. Wisotzky, Jennifer Semel, William S. Weintraub.

**Methodology:** Suzanne M. Simkovich, Naheed Ahmed, Derek DeLia, William S. Weintraub.

**Project administration:** Suzanne M. Simkovich, Naheed Ahmed, Eric M. Wisotzky, Jennifer Semel, Kathryn Pellegrino.

**Resources:** Eric M. Wisotzky.

**Supervision:** Derek DeLia, William S. Weintraub.

**Writing – original draft:** Suzanne M. Simkovich, Naheed Ahmed, Asli McCullers.

**Writing – review & editing:** Suzanne M. Simkovich, Naheed Ahmed, Eric M. Wisotzky, Jennifer Semel, Kathryn Pellegrino, Derek DeLia, William S. Weintraub.

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
