## [Decision Letter · Decision Letter 0]

25 May 2022

PONE-D-22-03353Health, Social, and Economic Characteristics of Patients Enrolled in a COVID-19 Recovery ProgramPLOS ONE

Dear Dr. Simkovich,

Thank you for submitting your manuscript to PLOS ONE. After careful consideration, we feel that it has merit but does not fully meet PLOS ONE’s publication criteria as it currently stands. Therefore, we invite you to submit a revised version of the manuscript that addresses the points raised during the review process.

We look forward to receiving your revised manuscript.

Kind regards,

Jeffrey S. Hallam, Ph.D., FRSPH

Academic Editor

PLOS ONE

Journal Requirements:

2. Thank you for stating the following in the Acknowledgments/Funding Section of your manuscript: 

This project did not have assigned funding for its completion. Suzanne M. Simkovich was supported by funding from the National Heart, Lung, and Blood Institute K12HL137942.

This project did not have assigned funding for its completion. Suzanne M. Simkovich was supported by funding from the National Heart, Lung, and Blood Institute K12HL137942.The funders had no role in study design, data collection and analysis, decision to publish, or preparation of the manuscript.

Additional Editor Comments:

Manuscript PONE-D-22-03353 entitled "Health, Social, and Economic Characteristics of Patients Enrolled in a COVID-19 Recovery Program" which you submitted to PLOS ONE.

The reviewer(s) suggest major revisions to your manuscript. Therefore, I invite you to respond to the reviewer(s)' comments and revise your manuscript.

Once again, thank you for submitting your manuscript to PLOS ONE and I look forward to receiving your revision.

Sincerely,

Jeffrey S. Hallam, PhD, FRSPH

Academic Editor

Reviewers' comments:

Reviewer's Responses to Questions

**Comments to the Author**

1. Is the manuscript technically sound, and do the data support the conclusions?

Reviewer #1: Partly

Reviewer #2: Partly

2. Has the statistical analysis been performed appropriately and rigorously? 

Reviewer #1: Yes

Reviewer #2: No

3. Have the authors made all data underlying the findings in their manuscript fully available?

Reviewer #1: Yes

Reviewer #2: Yes

4. Is the manuscript presented in an intelligible fashion and written in standard English?

Reviewer #1: Yes

Reviewer #2: Yes

5. Review Comments to the Author

Reviewer #1: Thank you for the opportunity to review this interesting cross sectional study. The manuscript reports a diverse range of person-centred outcomes for attendees in a COVID-19 recovery programme who experienced ongoing symptoms. Attendees reported impaired quality of life, increased fatigue and a significant minority reported problems with meeting basic needs such as paying rent or being able to afford food. The authors highlight these ongoing impairments experienced by people after acute COVID-19 and the need for services to promote recovery.

This study has strengths including the diverse array of person-centred measures reported in the clinic. However, there are a number of points for consideration for the authors which I detail below.

1. A key limitation of the study is the convenience sampling strategy. As patients were recruited from a clinical service after identification of having ongoing symptoms, it is difficult for the reader to place the study population in some sort of context. This is a convenience sample, conditional on patients having symptoms. The timing of referral to the service in relation to acute infection unclear, and the representativeness of the study population is unclear. Could the authors provide some information on the wider population from which the study sample was drawn? For example, the demographics of the local community, and/or population level statistics on COVID-19 prevalence, % admitted to hospital/ICU, would be helpful. Regardless, the authors should explicitly state this limitation, highlighting that generalizability of results is not possible due to this sampling strategy.

2. The severity of COVID-19 is an important factor associated with ongoing symptoms. It is unclear if “Referral to Clinic” is a valid proxy for disease severity. This would be clear for readers if a separate variable is resented in the baseline characteristics table with disease severity defined by where the patient was managed: community, hospital, ICU.

3. It would be useful to see risk factors for severe COVID-19 presented in the main table.

4. It is unclear what the minimum clinically important difference (MCID) is for the outcomes presented. Is one standard deviation accepted to be the MCID? If so, I’d suggest explicitly stating this in the table which provides detail in relation to each outcome.

5. Analyses appear in results which are not described in methods - please ensure these are described in the methods section.

6. Ensuring that the manuscript is reported to STROBE standards would make reading it clearer.

7. I would suggest referencing relevant, large studies which have been published in relation to post-acute COVID-19 symptoms, such as PHOSP-COVID in the UK (e.g. Physical, cognitive, and mental health impacts of COVID-19 after hospitalisation (PHOSP-COVID): a UK multicentre, prospective cohort study. Lancet Respir Med. 2021 Nov;9(11):1275-1287. doi: 10.1016/S2213-2600(21)00383-0. Epub 2021 Oct 7. Erratum in: Lancet Respir Med. 2022 Jan;10(1):e9. PMID: 34627560; PMCID: PMC8497028.) and analyses using the VA database (e.g. Risks of mental health outcomes in people with covid-19: cohort study. BMJ. 2022 Feb 16;376:e068993. doi: 10.1136/bmj-2021-068993. PMID: 35172971; PMCID: PMC8847881.)

Reviewer #2: Recommend that authors include COVID-19 hospitalization and ICU data on one of the descriptive tables, as it is touched upon in narrative.

Disability is not an occupation, as shown on Table 1. The authors conflate this “occupation” with disability status (line 176). It does not appear that actual disability status was otherwise measured, which limits the findings of this study.

Authors should note in-text which version of WPAI was used.

The WPAI-GH reflects general health, and is not specific to symptoms caused by PACS

The WPAI results presented in Table 3 and discussed (lines 213-218) are not consistent with WPAI scoring. It appears the authors coded responses dichotomously, which invalidates the instrument.

There is no indication of the length of time between infection and measurements, which may influence results and thus, potentially, affects findings.

Age distribution is not provided. Further, no rationale was provided for the age 50 cut-off. Thus, reporting age-group related differences in PROMIS-29 scores seems superfluous.

Mean age was reported as 48 (line 42) and 45 (line 260)

The authors overreach in interpreting the data. For example, the SDOH questions are not directly linked to PACS, nor are they compared to any referent, yet the authors suggest a relationship. Given the study population is largely “young and employed” (line 243), the social needs and financial hardships noted (lines 234-5) are an indictment of our current economic situation than PACS related. Additionally, “increased fatigue” (line 235) implies causation.

A limitation that is not addressed is that the surveys/instruments were administered prior to their appointment, increasing the likelihood of social desirability bias.

Recommend that authors remove “suffers” from the text as it is ableist language.

6. PLOS authors have the option to publish the peer review history of their article (what does this mean?). If published, this will include your full peer review and any attached files.

Reviewer #1: No

Reviewer #2: No

---

## [Author Response · Author response to Decision Letter 0]

3 Aug 2022

Reviewer #1: Thank you for the opportunity to review this interesting cross sectional study. The manuscript reports a diverse range of person-centred outcomes for attendees in a COVID-19 recovery programme who experienced ongoing symptoms. Attendees reported impaired quality of life, increased fatigue and a significant minority reported problems with meeting basic needs such as paying rent or being able to afford food. The authors highlight these ongoing impairments experienced by people after acute COVID-19 and the need for services to promote recovery.

This study has strengths including the diverse array of person-centred measures reported in the clinic. However, there are a number of points for consideration for the authors which I detail below.

R1C1. A key limitation of the study is the convenience sampling strategy. As patients were recruited from a clinical service after identification of having ongoing symptoms, it is difficult for the reader to place the study population in some sort of context. This is a convenience sample, conditional on patients having symptoms. The timing of referral to the service in relation to acute infection unclear, and the representativeness of the study population is unclear. Could the authors provide some information on the wider population from which the study sample was drawn? For example, the demographics of the local community, and/or population level statistics on COVID-19 prevalence, % admitted to hospital/ICU, would be helpful. Regardless, the authors should explicitly state this limitation, highlighting that generalizability of results is not possible due to this sampling strategy.

R1A1. We agree with your suggestion to enhance the context of the population enrolled in our program. To do this, we have added a more thorough description of the program (MSHCRP) along with a description of community served by the program (ln 83-91). Second, we have added a clear description to the methods for referral to the clinic (lines 87-90) and added that patients do not have to previously be affiliated with MedStar to come to this program (ln 90-91. Third, we added descriptive statistic -”time since COVID related symptoms began” to Table 1. The self- selection bias of this population is listed as the first limitation in the discussion as we recognize the patients enrolled in our program have chosen to engage in our program and have to recognize ongoing symptoms (ln 268-269). Further, in the discussion, we explained how our cohort is different from other studies as we only included those who were referred or self-referred for rehabilitation not all patients who had COVID-19 (ln 242-258). 

R2C2. The severity of COVID-19 is an important factor associated with ongoing symptoms. It is unclear if “Referral to Clinic” is a valid proxy for disease severity. This would be clear for readers if a separate variable is resented in the baseline characteristics table with disease severity defined by where the patient was managed: community, hospital, ICU.

R1A2. We have edited Table 1 to show the number of patients who were managed in the hospital, ICU and required a ventilator. This is currently our best marker of severity for our population. We agree that disease severity is not based on “referral to clinic.” Based on the current literature, per United States Centers for Disease Control Guidance, severe COVID-19 would require hospitalization and is our best surrogate marker collected. We have placed the reference for the United States Centers for Disease Control Guidance for severe COVID-19 as reference 32 and has been added to line 236. 

R1C3. It would be useful to see risk factors for severe COVID-19 presented in the main table.

R1A3. We have edited Table 1 to show the number of patients who were managed in the hospital, ICU and the that required a ventilator. This is currently our best marker of severity for our population. We agree that disease severity is not based on “referral to clinic.” Based on the current literature, per United States Centers for Disease Control Guidance, severe COVID-19 would require hospitalization and is our best surrogate marker collected. We have placed the reference for the United States Centers for Disease Control Guidance for severe COVID-19 as reference 32 and has been added to line 236. 

R1C4. It is unclear what the minimum clinically important difference (MCID) is for the outcomes presented. Is one standard deviation accepted to be the MCID? If so, I’d suggest explicitly stating this in the table which provides detail in relation to each outcome.

R1A4. We chose not to state the MCID as we are not comparing two distinct groups but want to show the range of what is considered a score for the general population by utilizing the mean and standard deviation. We note when our cohort does not fall within this range. 

R1C5. Analyses appear in results which are not described in methods - please ensure these are described in the methods section.

R1A5. We have ensured all analyses are stated in the methods. We have clarified the following points in the methods: 

- Added the following text “We conducted a subgroup analysis by age (≥ 50 years or < 50 years), race/ethnicity (Asian, Black, Hispanic, White, unknown or other), gender (male or female), medical diagnosis of asthma (yes or no) or allergies (yes or no) or diabetes (yes or no) and hospitalized for COVID-19 (yes or no) in relation to PROMIS scores." (ln 150-153)

- Added the following text “All surveys were scored in accordance with the parameters of the survey.” (ln 146-147)

R1C6. Ensuring that the manuscript is reported to STROBE standards would make reading it clearer.

R1A6. We have edited the manuscript to report our findings according to STROBE standards. We have added a line reflecting this to the methods (ln 106-108) and we have added the STROBE standards checklist to the supplement with page numbers of items. 

R1C7. I would suggest referencing relevant, large studies which have been published in relation to post-acute COVID-19 symptoms, such as PHOSP-COVID in the UK (e.g. Physical, cognitive, and mental health impacts of COVID-19 after hospitalisation (PHOSP-COVID): a UK multicentre, prospective cohort study. Lancet Respir Med. 2021 Nov;9(11):1275-1287. doi: 10.1016/S2213-2600(21)00383-0. Epub 2021 Oct 7. Erratum in: Lancet Respir Med. 2022 Jan;10(1):e9. PMID: 34627560; PMCID: PMC8497028.) and analyses using the VA database (e.g. Risks of mental health outcomes in people with covid-19: cohort study. BMJ. 2022 Feb 16;376:e068993. doi: 10.1136/bmj-2021-068993. PMID: 35172971; PMCID: PMC8847881.)

R1A7. We added Evans et al as a citation to ln 302. We have excluded Xie et al as this study shows patients who suffer from COVID-19 have increased incidence of mental health disorders. Although this is an important study, we have established in our manuscript that PASC encompasses mental health disorders, and the goal of this manuscript is to focus on quality-of-life and socioeconomic status. 

Reviewer #2: 

R2C1. Recommend that authors include COVID-19 hospitalization and ICU data on one of the descriptive tables, as it is touched upon in narrative.

R2A1. We have added hospitalization, ventilator and ICU data to Table 1. 

R2C2. Disability is not an occupation, as shown on Table 1. The authors conflate this “occupation” with disability status (line 176). It does not appear that actual disability status was otherwise measured, which limits the findings of this study.

R2A2. We have removed disabled from the occupation list. The disabled population is encompassed in the “unemployed” group. This is a limitation that we do not know the actual disability status of this group. To address this, we have added the following to the limitations section in the discussion: “Lastly, we are assessing ability to work in our population, but we do not know the true disability status of our patients which may impact the ability to work” (ln 281-283).

R2C3. Authors should note in-text which version of WPAI was used.

R2A3. We have added to the text that the SHP version of the WPAI was used (ln 135). 

R2C4. The WPAI-GH reflects general health, and is not specific to symptoms caused by PACS

R2A4. We utilized the WPAI-SHP. We have clarified this in the methods (ln 135). When the survey (available in the online supplement) was administered, COVID-19, was replaced as the disease affecting work and productivity. The full survey administered is available in the online supplement. 

R2C5. The WPAI results presented in Table 3 and discussed (lines 213-218) are not consistent with WPAI scoring. It appears the authors coded responses dichotomously, which invalidates the instrument.

R2A5. Responses are not coded dichotomously. Scoring was performed according to the algorithm provided by Reilley and associates accessible at the following website (http://www.reillyassociates.net/wpai_scoring.html), which is listed in the references (#20). We have removed the statement “Therefore, over 50% of the cohort reported significant impairment in work productivity in three of the four areas assessed.” 

R2C6. There is no indication of the length of time between infection and measurements, which may influence results and thus, potentially, affects findings.

R2A6. We have added the length of time between infection in measurement to Table 1. Inclusion in this study required that a patient must be at least 6 weeks from the first day of symptom onset from COVID-19. This was clarified in the methods (ln 87-89). 

R2C7. Age distribution is not provided. Further, no rationale was provided for the age 50 cut-off. Thus, reporting age-group related differences in PROMIS-29 scores seems superfluous. Mean age was reported as 48 (line 42) and 45 (line 260)

R2A7. We have reported an age distribution in the text by providing the standard deviation of the mean age in Table 1. Age selection of 50 years was selected as prognostic of clinical outcomes as the literature supports worse outcomes with increased age for patients who have COVID-19. The following article was added as a reference to support this selection (reference #21):

Zheng Z, Peng F, Xu B, Zhao J, Liu H, Peng J, Li Q, Jiang C, Zhou Y, Liu S, Ye C, Zhang P, Xing Y, Guo H, Tang W. Risk factors of critical & mortal COVID-19 cases: A systematic literature review and meta-analysis. J Infect. 2020 Aug;81(2):e16-e25. doi: 10.1016/j.jinf.2020.04.021. Epub 2020 Apr 23. PMID: 32335169; PMCID: PMC7177098.

R2C8. The authors overreach in interpreting the data. For example, the SDOH questions are not directly linked to PACS, nor are they compared to any referent, yet the authors suggest a relationship. Given the study population is largely “young and employed” (line 243), the social needs and financial hardships noted (lines 234-5) are an indictment of our current economic situation than PACS related. Additionally, “increased fatigue” (line 235) implies causation.

R2A8. We agree that we cannot directly associate SDOH questions with PASC. We do not agree though that SDOH questions are only indicative of our current economic situation. The “young and employed” do not necessarily reflect the current economic situation. SDOH could be based on multiple factors. To clarify this important point, we have added the following language “Based on the limits of this investigation, we can not conclude this is due to PASC, but indicates or cohort has social and financial impairments” (ln 238-239). 

R2C9. A limitation that is not addressed is that the surveys/instruments were administered prior to their appointment, increasing the likelihood of social desirability bias.

R2A9. We agree that this is a limitation to the analysis. We have added the following lines to address this limitation: “We administered surveys prior to the first visit in the clinic and responses are self-reported, thus creating a potential social desirability bias” (ln 268-270).

R2C10. Recommend that authors remove “suffers” from the text as it is ableist language.

R2A10. We have removed the term “suffers” from the text.

---

## [Decision Letter · Decision Letter 1]

11 Nov 2022

Health, Social, and Economic Characteristics of Patients Enrolled in a COVID-19 Recovery Program

PONE-D-22-03353R1

Dear Dr. Simkovich,

We’re pleased to inform you that your manuscript has been judged scientifically suitable for publication and will be formally accepted for publication once it meets all outstanding technical requirements.

Kind regards,

Jeffrey S. Hallam, Ph.D., FRSPH

Academic Editor

PLOS ONE

Additional Editor Comments (optional):

Reviewers' comments:

Reviewer's Responses to Questions

**Comments to the Author**

1. If the authors have adequately addressed your comments raised in a previous round of review and you feel that this manuscript is now acceptable for publication, you may indicate that here to bypass the “Comments to the Author” section, enter your conflict of interest statement in the “Confidential to Editor” section, and submit your "Accept" recommendation.

Reviewer #1: All comments have been addressed

Reviewer #2: All comments have been addressed

2. Is the manuscript technically sound, and do the data support the conclusions?

Reviewer #1: Yes

Reviewer #2: Yes

3. Has the statistical analysis been performed appropriately and rigorously? 

Reviewer #1: Yes

Reviewer #2: Yes

4. Have the authors made all data underlying the findings in their manuscript fully available?

Reviewer #1: Yes

Reviewer #2: Yes

5. Is the manuscript presented in an intelligible fashion and written in standard English?

Reviewer #1: Yes

Reviewer #2: Yes

6. Review Comments to the Author

Reviewer #1: The authors have comprehensively responded to my comments.

Reviewer #2: (No Response)

7. PLOS authors have the option to publish the peer review history of their article (what does this mean?). If published, this will include your full peer review and any attached files.

Reviewer #1: No

Reviewer #2: No

---

## [Editor Report · Acceptance letter]

18 Nov 2022

PONE-D-22-03353R1 

Health, Social, and Economic Characteristics of Patients Enrolled in a COVID-19 Recovery Program 

Dear Dr. Simkovich:

I'm pleased to inform you that your manuscript has been deemed suitable for publication in PLOS ONE. Congratulations! Your manuscript is now with our production department. 

Kind regards, 

on behalf of

Dr. Jeffrey S. Hallam 

Academic Editor

PLOS ONE